# Andalusian Organic Farming Plans (2002–2016): Themes, Approaches and Values

José-Francisco Jiménez-Díaz [1,*] and Francisco Collado-Campaña [2]

1 Department of Public Law, Universidad Pablo de Olavide, 41013 Sevilla, Spain
2 Department of Political Science, Public International Law and Procedural Law, Universidad de Málaga, 29016 Málaga, Spain; fcolcam@uma.es
* Correspondence: josefco@upo.es

**Abstract:** Organic farming in the Spanish region of Andalusia has acquired great socio-economic importance over the past decades. The purpose of this article is to study the themes, approaches, and socio-political values pertaining to ecological agriculture addressed in the Andalusian plans for organic farming (2002–2016). The contents of these plans have not been systematically studied before. From a descriptive and qualitative perspective, the authors present and classify the main themes addressed in those plans and show the socio-political approaches and values that underpin the plans. A thematic and semantic content analysis methodology is applied to the plans and sections addressing various objectives, measures, and actions. A theoretical-qualitative sampling is developed, and 109 keywords are selected for content analysis. This analysis allows us to detect numerous themes pertaining to ecological agriculture and to classify them into six semantic fields linked to various approaches and values promoted by the Andalusian autonomous administration. Therefore, this research focuses on changing perspectives of organic farming developed by the administration and the agents involved in the plans. The authors conclude that the diverse actors have prioritized a productivist–technocratic approach to ecological agriculture, to the detriment of an approach centered on sustainable and agroecological local communities.

**Keywords:** Andalusia; Andalusian plans for organic farming; content analysis; ecological agriculture; organic farming approaches; socio-political values



## 1. Introduction

Andalusia is the most populated Spanish autonomous community, with the greatest diversity of natural and agrarian landscapes, and with the largest conventional and ecological agricultural production. Specifically, the certified organic agricultural area of Andalusia exceeded one million hectares in 2018, concentrating 46.8% of the land area dedicated to organic farming in Spain. Andalusia also stands out both in the production and in the transformation industry of organic products, since it brings together the highest proportion of producer operators in Spain with more than a third of the national total [1]. In other words, Andalusia is a benchmark in ecological agriculture in Europe. In addition, agricultural activity is a strategic sector in the Andalusian economic structure in several aspects: It generates employment and wealth, contributes to cohesion and balance between territories, and "helps to establish population in rural areas" [2].

For its part, for more than three decades, the autonomous community of Andalusia has been endowed with self-government capacity and, therefore, has its own governmental structure (Junta de Andalucía, autonomous government), as well as a legislative power represented in the Andalusian Parliament. Both autonomous powers (executive and legislative) have allowed the Junta de Andalucía, since its constitution in 1983, to develop its own public policies in a broad range of spheres. Furthermore, the Junta de Andalucía has exclusive competence in development and execution in certain matters. This is the case of the competence on "agriculture, livestock and rural development" [3] (Art. 48.1 of

the Statute of Autonomy of Andalusia) and, more specifically, on "management, planning, reform and development of the agricultural and livestock sectors and agri-food and, especially, the improvement and management of agricultural, livestock and agroforestry farms. Regulation of agricultural production processes [ ... ] Ecological agriculture, food sufficiency, and technological innovations" [3] (Art. 48.3a). In representative democracies, government and administrative action corresponds not only to the formal institutions of the central government (state executive power), but also to political entities located at the lower levels of the State endowed with a margin of autonomy and of political initiative, as  is the case of the autonomous communities in Spain. Therefore, it is necessary to know the actions of these other levels of government in the administration of public affairs. In this regard, Andalusian plans for organic farming, developed in the autonomous region of Andalusia, are analyzed.

Undoubtedly, the production of quality organic food in a traditionally agricultural region such as Andalusia, which exceeds 8.4 million inhabitants [4] and with a land area of 87,589.90 km$^2$, needs to be studied in some detail and from various perspectives [5]. It is thus necessary to consider the visions of the multiple agents that participate in the elaboration of public policies on agriculture and/or ecological production: farmers, experts and technicians, environmental associations, political groups, producer associations, trade unions, marketing agents, agri-food distributors, etc. In fact, the I Andalusian Plan for Ecological Agriculture (*Plan Andaluz de la Agricultura Ecológica*, I-PAAE) [6] was drawn up through dialogue with and contributions and agreements from the main socio-economic, political, and institutional actors involved in the organic farming sector. Therefore, in this first plan and in the two following plans, II Andalusian Plan for Ecological Agriculture 2007–2013 (*II Plan Andaluz de Agricultura Ecológica 2007–2013*, II-PAAE) [7] and III Andalusian Plan for Ecological Production (*III Plan Andaluz de la Producción Ecológica*, III-PAPE) [8], the sociopolitical values that were implicit or explicit in them and that were the result of the negotiations and agreements of the aforementioned actors in a changing social context can be analyzed. As this context has changed since the beginning of this century—in 2002 the I-PAAE was published, and in 2016 the III-PAPE was released—it is convenient to investigate the possible evolution of the main themes, approaches, and values related to ecological agriculture in the referred plans. In fact, there was a change of name between the first and third organic farming plans: from "organic farming" to "organic production." This indicates that there were significant changes in the conception and vision of the ecological rural environment in the plans.

This article addresses the following research question: How have the main themes, approaches, and socio-political values associated with ecological agriculture evolved in the three Andalusian plans for organic farming? These plans, published by the Andalusian autonomous administration between 2002 and 2016, have not been studied, until now, from the perspective of a systematic analysis of their contents. However, the main themes highlighted in the plans could condition the prioritized approaches and values in the political planning of organic farming in Andalusia. Those themes could also influence the practices of Andalusian organic farmers. As mentioned above, Andalusian organic farming has acquired great socio-economic importance over the past decades, since Andalusia is the Spanish region with the largest area of land dedicated to this agricultural activity. Consequently, this investigation contributes to the detailed understanding of the themes, approaches and socio-political values underlying these plans.

Two exploratory hypotheses serve as a starting point. First, if the context of Andalusian agriculture and the political and socioeconomic interests of regional actors, between 2002 and 2016, have experimented important changes, then the themes, approaches, and socio-political values associated with ecological agriculture have varied substantially in the three Andalusian plans [6–8]. Second, if these plans have been conditioned by the changes in the context of Andalusian agriculture and by various interests involved, then these plans are oriented towards a conventionalized and instrumental vision of ecological rural environment.

Indeed, these variations are caused by a resignification and/or redefinition of what is understood by organic farming in every situation and context. Ecological agriculture becomes in this manner an empty signifier and, to a certain extent, a semantic and symbolic battlefield in which the agents involved in the definition and implementation of public policies on ecological agriculture in Andalusia engage in certain symbolic struggles to always prioritize the approaches and values that are most consistent with their respective situations, contexts, and political and socio-economic interests. Therefore, the two-dimensional perspective of political power is analyzed, understood as the ability to define and control the issues on the political agenda, as well as "decision-making" and "non-decision making" [9]. In this regard, the themes, approaches, and values of ecological agriculture that have been prioritized and neglected in this agenda are studied.

For their part, socio-political values are conceived here as "the criteria of preference and latitudes of acceptance, or rejection, attributed to the ideas and facts that guide the behavior of social actors in their courses of action" [10] (p. 27) [11]. In other words, values are "ideas that individuals or human groups hold about what is desirable, appropriate, good or bad [ . . . ] The specific culture in which individuals spend their lives strongly influences what they value" [12] (p. 770) [13]. Specifically, in this work, the values show certain changes in the evaluations and/or perceptions of the agents who devised the three plans for organic farming in Andalusia, especially as a function of the changes produced in the European and Andalusian regulatory framework on said agricultural activity between 2002 and 2016. Likewise, significant changes are observed in the definition and assessment of ecological agriculture in the plans.

Therefore, in this article, using content analysis methodology applied to Andalusian plans for organic farming—all those available up to this moment—and through the qualitative–theoretical sampling strategy, explained later, we show various types of social values linked to organic farming. Specifically, social values are conceived as the ideas and criteria of preference that are internalized by individuals or human groups throughout their socialization process, with the former functioning as cognitive mechanisms to judge the social world and the natural environment in which these individuals are immersed. For its part, agriculture depends on the social processes of production and reproduction that mainly take place in the rural context and environment, as well as on the local and natural resources available in this environment. Furthermore, in the past decades, agriculture has experienced a growing relationship and interdependence with other actors and contexts, due to the increasing globalization, industrialization, modernization, and urbanization of social life [10–13].

Thus, organic farming, conceived as field of human activity, is subject to the daily evaluations of the individuals who carry out this agricultural practice, as well as to the evaluations of the socio-political agents who try to administer, regulate, and intervene in this practice. Indeed, different assessments of organic farming have spread throughout the Western world, particularly in Europe, during recent times [1,2,6–8,10]. In summary, social values allow individuals and agents to discriminate or differentiate between what they consider good or bad, appropriate, or inappropriate about the ideas and practices associated with ecological agriculture. In this sense, it is necessary to study in detail the values displayed by those who have intervened in or collaborated on the elaboration of Andalusian plans for organic farming. In this way, as the authors expose in Section 4.2, six sets of social values linked to the two general approaches to ecological agriculture stand out. On the one hand is the ecologist approach that encompasses values such as diversity, respect for the environment, food security, local and sustainable development, sustainability, gender equality, participation, and health. On the other hand is the productivist–technocratic approach, which emphasizes values such as competitiveness, rural development, innovation, growth, quality, cooperation, regulation, and control. The following pages provide arguments and data that show how the latter approach prevailed over the former in the organic farming plans studied, especially in the third plan (III-PAPE) [8].

Consequently, this article proposes two specific objectives: first, to expose and classify the main themes associated with ecological agriculture that guide the aforementioned plans, and second, to show the socio-political approaches and values on which the three Andalusian plans for organic farming are based. These approaches and values can change over time, especially based on the changing social positions of the agents and the transformations produced in the context. This is marked by the normative and regulatory frameworks of the organic farming sector in Andalusia, as well as by the evolution of organic farming in Spain and the European Union in the scenario of increasingly globalized and interdependent societies during the first two decades of the present century. To fulfill both objectives, the article is structured in five sections, including this introductory section. The second section develops the theoretical foundations of the research, as well brief methodological notes. The third section is dedicated to the justification of the methodology. The fourth section shows, on the one hand, the context of Andalusian ecological agriculture, and on the other, the analysis and classification of the themes, approaches, and values displayed in the plans studied. Finally, the discussion, conclusions, new research questions, limitations, and guidelines for future research are presented in the fifth section.

## 2. Theoretical Foundations

There are currently different definitions or approaches concerning organic production and/or organic farming. In this way, various visions are developed on the role that this activity and agrarian context should have in the change of role conventionally assigned to the farmer, as well as in the transformation of the relationships between agricultural activity, rural environment, development, growth, nature, territory, and food [14,15] [16] (p. 2). Those visions can be grouped into two general and opposing approaches that opt for different socio-political values concerning ecological agriculture: on the one hand, a holistic vision of ecological agriculture that can be subsumed under so-called agroecology, and on the other, a productivist and technocratic vision usually preferred by public administrations. Thus, at the beginning of the first Andalusian Plan for Ecological Agriculture (I-PAAE), it was said that,

"For certain sectors of society, organic farming is not merely a particular mode of production or processing of certain products; rather, it is incorporated into a broader concept called Agroecology [17]. This concept appears as a new paradigm of knowledge, conceived as a theoretical and methodological approach to study agricultural activity from an ecological perspective, jointly analyzing all the elements of agricultural processes: mineral cycles, energy transformations, biological processes and socioeconomic relationships" [6] (p. 17).

Likewise, within the paradigm of agroecology, four general conceptions can be identified of the processes and dynamics of the agroecological social transition defended from this point of view [18] (pp. 254–255): first, the eco-structural approach, centered on sustainable production and social metabolism with a macro perspective [19–22]; second, the social innovations model, focused on cooperative and endogenous tools with a micro approach [23–26]; third, the personal and collective agency model, concerned with micro and meso networks with a social institutional approach [27–29]; and, fourth, the post-development approach, oriented towards social change and food sovereignty with a macro approach [30–33].

However, for the public administrations that elaborate on definitions of organic farming, its conceptualization is much more instrumental and technocratic, opting for a rational–technical approach that avoids socioeconomic relations, the political context, and the ecosystems where said agricultural practice develops. In this sense, the first paragraph of the introduction of the I-PAAE was a declaration of intent and implicitly a commitment to certain socio-political values on ecological agriculture. Said paragraph set out the particular technocratic vision prioritized in the elaboration and development of the three Andalusian plans for organic farming and that therefore has conditioned their objectives, actions, and measures [34,35], as is shown in this article:

"Regulation (EEC) No. 2092/91 on organic agricultural production and its indication in agricultural and food products, considers organic agriculture as that which complies with basic principles of production included in Annexes I and III. Annex II contains the list of phytosanitary products, detergents, fertilizers or soil conditioners that can be used, and may only be used under the specific conditions set forth in Annexes I and II and to the extent that the corresponding use is authorized in the general agriculture of the Member State" [6] (p. 17).

To make clear this vision of ecological agriculture supported by the I-PAAE, at the end of a very brief disquisition on the possible approaches the technocratic vision mentioned earlier is chosen: "Although broader conceptions are also valid, for the purposes of this Plan we will only consider Organic Agriculture, what is in compliance with the provisions of Regulation 2092/91 cited at the beginning of this Plan" [6] (p. 17).

Consequently, these two general visions compete for the definition of ecological agriculture, and each of them implies certain commitments and/or priorities regarding certain socio-political values that are difficult to reconcile in the current socio-economic contexts of highly complex, interdependent, and globalized risk societies [36,37]. Undoubtedly, such general visions are based on the ideology of environmentalism whose "ultimate goal [is] the achievement of a sustainable society [ . . . ] that implies a series of changes that affect politics, the economy and the daily life of citizens" [38] (p. 734). This ideology adopts at least two different perspectives: one "ecologist" and another "environmentalist." Whereas this last perspective opts for an administrative approach to environmental problems, admitting that these can be solved without altering the present values of consumerism and productivism, the ecological perspective implies profound changes in the non-human natural world and in the form of social coexistence [38] (p. 730) [39]. Furthermore, the complex relationships between growth and development, and between environment and society [40], as well as the changes, problems, and challenges of rurality and European agricultural policy [14,41], cannot be ignored.

For its part, the International Federation of Organic Agriculture Movements (IFOAM) developed the vision of agroecology early on, conceiving organic farming as a "holistic system" that fosters the following socio-political values: "sustainable ecosystems, safe food, good nutrition, animal welfare and social justice" [16] (p. 2) [34] (pp. 43–45). IFOAM is a global organization that brings together social movements fighting in favor of organic farming, representing nearly 800 affiliated movements in 117 countries. This organization declares that its mission is "to lead, unite and help the ecological movement in all its diversity," and its vision is the "global adoption of ecologically, socially and economically sound systems, based on the principles of organic agriculture" [42]. This vision of ecological agriculture encompasses multiple dimensions, as it highlights "the role of this activity in obtaining ecologically healthy and self-sustainable agro-ecosystems (...), increasingly necessary in the face of the advance of climate change and the worsening of food crises" [16] (p. 2). For this reason, this vision advocates multi-activity and income diversification, the strategy of reducing inputs, production for sale in the local market, and local commerce. With this, an ecologically responsible development model is envisaged based on social justice as well as fostering greater autonomy of farmers in the face of the predominance of multinationals and agri-food companies [25,28].

The ecologist perspective of ecological agriculture (agroecology) is associated, according to Sevilla-Guzmán [31,43], with the traditional practices, uses, and perceptions incorporated by peasant societies, as well as with the transition towards a more sustainable social metabolism and towards other perspectives and socio-economic practices that can face the challenges of climate change and environmental deterioration [20,33] [21] (p. 82–83). This has implied considering ecological production a social action tending towards re-peasantization [16] (p. 2) [25,28]. This leads to "the reconfiguration of rural spaces into peasant spaces," that is, to the "strengthening of the peasant presence in the territory" [44] (p. 283) through the participation of agroecological social movements. In addition, re-peasantization is an "extensive and complex transition process, not yet com-

pleted, that unfolds along different dimensions, and is found at various levels of mutual interaction" [25] (p. 226), which seeks social sustainability through the autonomous reappropriation of the earth's resources, the increase in the added value of products, and the link between agriculture and society.

The environmentalist vision of ecological agriculture is shown in the technocratic definitions deployed by public administrations in their multiple regulations. In fact, the administrations opt for brief and operative definitions that allow for the regulation and control of the practice of organic farming. This is clear in the notion that states "that the first guidelines for the regulation of this activity in Europe defined organic farming as a set of agricultural techniques that exclude the use of synthetic chemicals" [16] (p. 2). This conception of ecological agriculture proposes a short-term and "negative definition of ecological production," highlighting "the aspects or practices that it eliminates or reduces, and for not taking into account issues related to soil fertility, the promotion of biodiversity, water treatment or the use of energy" [16] (p. 2) [45]. Thus, this way of conceiving organic farming is restricted to the possibility of replacing chemical and synthetic inputs "with other [inputs] certified as 'organic' that are more expensive, which perpetuates the dependence of farmers on the agrochemical companies that control this market" [16] (p. 2) [46]. Therefore, the success of this conceptualization is based "on the synthetic and informative definition that it proposes, as well as on its adaptation to current market requirements by not questioning the atomistic, rationalist and mechanistic approach of the industrialized agrarian model" [16] (p. 2) [17].

Consequently, in a scenario in which different ideological and political views are confronted about what is conceived and understood by ecological agriculture, the discourses of the socio-economic and political–institutional subjects become relevant because it is "in this plane in which the actors explain and argue their ways of acting, their desires, expectations and motivations" [16] (p. 3) [47,48]. This article contributes to identifying both visions, already objectified and reified, of the agents who participated in the elaboration of the mentioned Andalusian plans for organic farming. These, without a doubt, have largely conditioned the evolution of this agricultural practice in Andalusia during the past decades [21,34,35,49,50].

Based on the previous theoretical arguments and concerning the information and data on the context of Andalusian organic agriculture (see Section 4.1), this article applies thematic and semantic content analysis, as explained in the following section. The empirical research is based on a qualitative sample of 109 keywords in the form of nouns (see Appendix A, Table A1) linked to the general approaches to organic farming and, in turn, related to the contextualization presented of the Andalusian ecological rural environment. In this sense the nouns and themes that illustrate such approaches are selected and explicitly show different socio-political values associated with Andalusian ecological agriculture. Therefore, the qualitative–theoretical sampling strategy is used in the selection of nouns to establish a feedback between the theoretical approaches of organic farming and the reality of ecological agriculture in Andalusia. In other words, theoretical sampling allows, on the one hand, the generation of interpretations from which data are collected, encoded, and analyzed, and on the other, the development of criteria to decide which data to collect and where to find them, and the development of an interpretation that is more adjusted to the empirical reality studied [51–54].

The selected nouns become keywords or fundamental linguistic tools for the detailed thematic analysis of the exposed themes in the three organic farming plans studied. Indeed, these nouns are associated with the keywords selected for the analysis of the themes of ecological agriculture, included in the plans. The content analysis method is applied in several phases: On the one hand, all the documents of Andalusian plans for organic farming are analyzed to describe how the keywords evolve based on their respective absolute and relative frequencies (see Tables 1–3). On the other, the most relevant sections of these documents (objectives, actions, and SWOT –strengths, weaknesses, opportunities, and threats– analysis) are analyzed to check the logic of the quantitative evolution of the

keywords (see Table 3). This, in turn, allows a qualitative and semantic content analysis to be carried out in which the numerous keywords are categorized in six semantic fields linked to different socio-political approaches and values of organic farming in Andalusia (see Table 2). In other words, these semantic fields allow for the analysis and comparison of the meanings acquired by the themes on ecological agriculture, by means of the keywords, for the agents who participated in the elaboration of the three analyzed plans. However, these agents were not interviewed to find out the specific meanings they attribute to the themes, values and approaches studied, as this would require a different investigation.

**Table 1.** Keywords in Andalusian plans for organic farming: absolute and relative frequencies (the latter in parentheses).

| Key Words | I-PAAE, 2002: I Andalusian Plan for Ecological Agriculture | II-PAAE, 2007: II Andalusian Plan for Ecological Agriculture | III-PAPE, 2016: III Andalusian Plan for Ecological Production |
|---|---|---|---|
| Administration(s) | 19 (0.52) | 35 (1.42) | 19 (0.78) |
| Organic farming | 681 (18.65) | 222 (9.04) | 26 (1.06) |
| Agroecology | 11 (0.30) | 2 (0.08) | 3 (0.12) |
| Agroindustry(ies) | 2 (0.05) | 12 (0.49) | 26 (1.06) |
| Water | 7 (0.19) | 4 (0.16) | 0 (0) |
| Food | 42 (1.15) | 18 (0.73) | 18 (0.73) |
| Organic food | 14 (0.38) | 48 (1.95) | 31 (1.27) |
| Counseling | 12 (0.33) | 45 (1.83) | 35 (1.43) |
| Grant(s) | 174 (4.76) | 86 (3.50) | 29 (1.18) |
| Public aid | 4 (0.11) | 0 (0) | 1 (0.04) |
| Beneficiary(ies) | 5 (0.14) | 2 (0.08) | 0 (0) |
| Benefit(s) | 0 (0) | 21 (0.85) | 11 (0.45) |
| Animal welfare | 3 (0.08) | 0 (0) | 4 (0.16) |
| Biodiversity | 0 (0) | 14 (0.57) | 19 (0.78) |
| Short supply chains | 0 (0) | 0 (0) | 8 (0.33) |
| Quality | 55 (1.51) | 53 (2.15) | 31 (1.27) |
| Climate change | 0 (0) | 8 (0.32) | 9 (0.37) |
| Certification | 55 (1.51) | 53 (2.15) | 18 (0.73) |
| Citizenry | 0 (0) | 0 (0) | 7 (0.29) |
| Commercialization | 63 (1.72) | 48 (1.95) | 54 (2.21) |
| Competitiveness | 2 (0.05) | 1 (0.04) | 21 (0.86) |
| Communication | 5 (0.14) | 4 (0.16) | 4 (0.16) |
| Concentration | 15 (0.41) | 8 (0.32) | 1 (0.04) |
| Supply concentration | 5 (0.14) | 6 (0.24) | 1 (0.04) |
| Knowledge | 54 (1.48) | 40 (1.62) | 49 (2) |
| Traditional knowledge | 0 (0) | 0 (0) | 2 (0.08) |
| Conservation | 12 (0.33) | 11 (0.45) | 15 (0.61) |
| Consumer(s) | 126 (3.45) | 40 (1.62) | 24 (0.98) |
| Consumption | 57 (1.56) | 105 (4.27) | 59 (2.41) |
| Pollution | 3 (0.08) | 20 (0.81) | 3 (0.12) |
| Control | 110 (3.01) | 45 (1.83) | 49 (2) |
| Cooperation | 15 (0.41) | 8 (0.32) | 19 (0.78) |
| Cooperatives | 18 (0.49) | 2 (0.08) | 5 (0.20) |
| Coordination | 14 (0.38) | 14 (0.57) | 17 (0.69) |
| Cost(s) | 15 (0.41) | 21 (0.85) | 8 (0.33) |
| Growth | 56 (1.53) | 22 (0.89) | 16 (0.65) |
| Development | 134 (3.67) | 166 (6.76) | 184 (7.52) |
| Local development | 0 (0) | 0 (0) | 13 (0.53) |
| Rural development | 12 (0.33) | 33 (1.34) | 43 (1.76) |
| Sustainable development | 1 (0.03) | 1 (0.04) | 12 (0.49) |
| Unemployed | 1 (0.03) | 0 (0) | 1 (0.04) |
| Gender inequalities | 0 (0) | 1 (0.04) | 1 (0.04) |

**Table 1.** *Cont.*

| Key Words | I-PAAE, 2002: I Andalusian Plan for Ecological Agriculture | II-PAAE, 2007: II Andalusian Plan for Ecological Agriculture | III-PAPE, 2016: III Andalusian Plan for Ecological Production |
|---|---|---|---|
| Diffusion | 25 (0.68) | 17 (0.69) | 21 (0.86) |
| Distribution | 70 (1.92) | 7 (0.28) | 24 (0.98) |
| Diversity | 10 (0.27) | 9 (0.37) | 1 (0.04) |
| Dissemination | 17 (0.46) | 4 (0.16) | 7 (0.29) |
| Education | 4 (0.11) | 8 (0.32) | 9 (0.37) |
| Employment | 10 (0.27) | 25 (1.02) | 23 (0.94) |
| Company(ies) | 39 (1.07) | 41 (1.67) | 40 (1.63) |
| Strategy(ies) | 12 (0.33) | 10 (0.41) | 34 (1.39) |
| Local development strategies | 0 (0) | 0 (0) | 13 (0.53) |
| Europe | 49 (1.34) | 2 (0.08) | 3 (0.12) |
| Experience(s) | 27 (0.74) | 6 (0.24) | 11 (0.45) |
| Training | 169 (4.63) | 60 (2.44) | 52 (2.12) |
| Gender | 0 (0) | 17 (0.69) | 18 (0.73) |
| Management | 22 (0.60) | 16 (0.65) | 26 (1.06) |
| Large retail | 15 (0.41) | 0 (0) | 4 (0.16) |
| Rural Development Groups | 1 (0.03) | 1 (0.04) | 12 (0.49) |
| Gender equality | 0 (0) | 2 (0.08) | 6 (0.24) |
| Environmental impact | 2 (0.05) | 2 (0.08) | 0 (0) |
| Social impact | 1 (0.03) | 4 (0.16) | 0 (0) |
| Incentive(s) | 1 (0.03) | 4 (0.16) | 63 (2.57) |
| Information | 45 (1.23) | 34 (1.38) | 47 (1.92) |
| Innovation | 3 (0.08) | 4 (0.16) | 17 (0.69) |
| Research | 159 (4.35) | 40 (1.63) | 30 (1.22) |
| Young people | 7 (0.19) | 8 (0.32) | 16 (0.65) |
| Environment | 28 (0.77) | 14 (0.57) | 27 (1.10) |
| Rural environment | 1 (0.03) | 10 (0.41) | 10 (0.41) |
| Domestic market | 6 (0.16) | 3 (0.12) | 7 (0.29) |
| Internal market | 5 (0.14) | 22 (0.89) | 2 (0.08) |
| Market(s) | 174 (4.76) | 57 (2.32) | 48 (1.96) |
| Woman/Women | 1 (0.03) | 37 (1.51) | 60 (2.45) |
| Norms | 24 (0.66) | 14 (0.57) | 9 (0.37) |
| Regulations | 21 (0.57) | 21 (0.85) | 28 (1.14) |
| Territorial planning | 7 (0.19) | 4 (0.16) | 23 (0.94) |
| Organizations | 36 (0.98) | 16 (0.65) | 17 (0.69) |
| Common Agricultural Policy | 5 (0.14) | 0 (0) | 3 (0.12) |
| Participation | 25 (0.68) | 27 (1.10) | 23 (0.94) |
| Participation of women | 0 (0) | 15 (0.61) | 9 (0.37) |
| Planning | 5 (0.14) | 11 (0.45) | 20 (0.82) |
| Policy(ies) | 19 (0.52) | 6 (0.24) | 12 (0.49) |
| Price(s) | 45 (1.23) | 24 (0.98) | 10 (0.41) |
| Organic production | 138 (3.78) | 194 (7.90) | 396 (16.18) |
| Productivity | 7 (0.19) | 1 (0.04) | 7 (0.29) |
| Producers | 107 (2.93) | 110 (4.48) | 20 (0.82) |
| Organic products | 202 (5.53) | 57 (2.32) | 69 (2.82) |
| Professional(s) | 23 (0.63) | 11 (0.45) | 37 (1.51) |
| Promotion | 50 (1.37) | 24 (0.98) | 34 (1.39) |
| Protection | 17 (0.46) | 17 (0.69) | 13 (0.53) |
| Natural resources | 1 (0.03) | 6 (0.24) | 4 (0.16) |
| Network(s) | 9 (0.25) | 19 (0.77) | 15 (0.61) |
| Regulation | 15 (0.41) | 6 (0.24) | 6 (0.24) |
| Profitability | 5 (0.14) | 3 (0.12) | 1 (0.04) |
| Residues | 7 (0.19) | 6 (0.24) | 6 (0.24) |
| Environmental protection | 3 (0.08) | 0 (0) | 1 (0.04) |
| Responsibility(ies) | 6 (0.16) | 1 (0.04) | 3 (0.12) |

**Table 1.** *Cont.*

| Key Words | I-PAAE, 2002: I Andalusian Plan for Ecological Agriculture | II-PAAE, 2007: II Andalusian Plan for Ecological Agriculture | III-PAPE, 2016: III Andalusian Plan for Ecological Production |
|---|---|---|---|
| Health | 15 (0.41) | 21 (0.85) | 8 (0.33) |
| Food safety | 11 (0.30) | 0 (0) | 1 (0.04) |
| Control system | 13 (0.35) | 11 (0.45) | 10 (0.41) |
| Certification schemes | 5 (0.14) | 5 (0.20) | 0 (0) |
| Sustainability | 4 (0.11) | 7 (0.28) | 15 (0.61) |
| Subsidy(ies) | 12 (0.33) | 22 (0.89) | 2 (0.08) |
| Mantainability | 0 (0) | 6 (0.24) | 1 (0.04) |
| Technique(s) | 32 (0.88) | 17 (0.69) | 28 (1.14) |
| Technology(ies) | 13 (0.35) | 24 (0.98) | 8 (0.33) |
| Territory(ies) | 3 (0.08) | 10 (0.41) | 32 (1.31) |
| Transfer | 15 (0.41) | 15 (0.61) | 14 (0.57) |
| Transformation | 46 (1.26) | 30 (1.22) | 22 (0.90) |
| Transparency | 1 (0.03) | 12 (0.49) | 13 (0.53) |
| Subtotal (words per column) | 3652 (100%) | 2456 (100%) | 2447 (100%) |

Source: Compiled by author. **Note**: The absolute frequencies of the keywords were recorded in each organic farming plan, and each of these frequencies were recorded independently. Each of the keywords was counted by itself in the three plans studied. The I Andalusian Plan for Ecological Agriculture [6] has an extension of 251 pages and 57,125 words, the II Andalusian Plan for Ecological Agriculture [6] runs to 123 pages and 32,171 words, and the III Andalusian Plan for Ecological Production [8] features 107 pages and 24,786 words. When deemed relevant, a search was made for the word in the singular and in the plural forms; for example, "water" and "waters" added up to 7 reiterations in the I Andalusian Plan for Ecological Agriculture; however, those same words did not appear at any time in the III Andalusian Plan for Ecological Production. Likewise, the relative frequencies—expressed in percentages and in parentheses— were calculated on the total number of words in each column, that is, on the total repetitions of all the keywords selected in each plan. The percentages were rounded to the second decimal.

**Table 2.** Approaches and values on ecological agriculture in the analyzed plans.

| Approaches in Andalusian Plans for Organic Farming | Semantic Field: Keywords | Values Associated with Semantic Field |
|---|---|---|
| **Illegitimate ecologist approach** | **Semantic field A:** water, agroecology, organic farming, diversity, environmental impact, social impact, environmental protection, food safety | Agroecology, diversity, respect for the environment, food safety |
| **Legitimate ecologist approach** | **Semantic field B:** organic food, animal welfare, biodiversity, short supply chains, climate change, citizenry, traditional knowledge, conservation, local development, sustainable development, gender inequalities, local development strategies, gender, Rural Development Groups, gender equality, woman/women, participation of women, natural resources, sustainability, maintainability | Local and sustainable development, gender equality, participation of women, sustainability |
| **Neutral ecologist approach** | **Semantic field C:** pollution, environment, domestic market, internal market, protection, participation, health | Protection, participation, health |
| **Legitimate productivist–technocratic approach** | **Semantic field D:** agroindustry(ies), counseling, benefit(s), competitiveness, commercialization, development, rural development, employment, company(ies), strategy(ies), management, incentive(s), information, innovation, young people, rural environment, regulations, territorial planning, planning, organic production, professional(s), territory(ies), transparency | Competitiveness, rural development, innovation, transparency |

**Table 2.** *Cont.*

| Approaches in Andalusian Plans for Organic Farming | Semantic Field: Keywords | Values Associated with Semantic Field |
|---|---|---|
| **Illegitimate productivist–technocratic approach** | **Semantic field E:** grant(s), public aid, beneficiary(ies), concentration, supply concentration, cost(s), consumer(s), cooperatives, growth, Europe, training, large retail, research, market(s), norms, Common Agricultural Policy (CAP), price(s), producers, profitability, certification schemes, subsidy(ies), transformation | Public aid, growth, Europe, profitability |
| **Neutral productivist–technocratic approach** | **Semantic field F:** administration(s), food, quality, certification, knowledge, control, consumption, communication, coordination, cooperation, unemployed, distribution, diffusion, dissemination, education, experience(s), organizations, policy(ies), promotion, network(s), regulation, residues, responsibility(ies), control system, technique(s), technology(ies), transfer | Quality, control, coordination, cooperation, regulation |

Source: compiled by author.

**Table 3.** Keywords in the Andalusian plans for organic farming in their sections on objectives, actions and/or interventions, and SWOT analysis.

| Keywords | I-PAAE, 2002I Andalusian Plan for Ecological Agriculture | II-PAAE, 2007II Andalusian Plan for Ecological Agriculture | III-PAPE, 2016III Andalusian Plan for Ecological Production |
|---|---|---|---|
| Administration(s) | 5 (0.90) | 6 (1.46) | 5 (1.16) |
| Organic farming | 34 (6.11) | 21 (5.10) | 3 (0.70) |
| Agroecology | 0 (0) | 0 (0) | 0 (0) |
| Agroindustry(ies) | 1 (0.18) | 3 (0.73) | 5 (1.16) |
| Water | 2 (0.36) | 0 (0) | 0 (0) |
| Food | 1 (0.18) | 1 (0.24) | 2 (0.47) |
| Organic food | 2 (0.36) | 13 (3.15) | 4 (0.93) |
| Counseling | 3 (0.54) | 6 (1.47) | 6 (1.40) |
| Grant(s) | 10 (1.80) | 7 (1.70) | 4 (0.93) |
| Public aid | 2 (0.36) | 0 (0) | 1 (0.23) |
| Beneficiary(ies) | 0 (0) | 0 (0) | 0 (0) |
| Benefit(s) | 0 (0) | 6 (1.46) | 0 (0) |
| Animal welfare | 0 (0) | 0 (0) | 0 (0) |
| Biodiversity | 0 (0) | 2 (0.48) | 1 (0.23) |
| Short supply chains | 0 (0) | 0 (0) | 3 (0.70) |
| Quality | 11 (1.98) | 6 (1.46) | 4 (0.93) |
| Climate change | 0 (0) | 2 (0.48) | 1 (0.23) |
| Certification | 8 (1.44) | 4 (0.97) | 3 (0.70) |
| Citizenry | 0 (0) | 0 (0) | 2 (0.47) |
| Commercialization | 9 (1.62) | 3 (0.73) | 16 (3.73) |
| Competitiveness | 0 (0) | 0 (0) | 5 (1.16) |
| Communication | 0 (0) | 0 (0) | 2 (0.47) |
| Concentration | 3 (0.54) | 1 (0.24) | 0 (0) |
| Supply concentration | 3 (0.54) | 1 (0.24) | 0 (0) |
| Knowledge | 19 (3.42) | 9 (2.18) | 8 (1.86) |
| Traditional knowledge | 0 (0) | 0 (0) | 1 (0.23) |
| Conservation | 1 (0.18) | 1 (0.24) | 0 (0) |
| Consumer(s) | 33 (5.93) | 2 (0.48) | 8 (1.86) |
| Consumption | 10 (1.80) | 24 (5.86) | 20 (4.66) |
| Pollution | 0 (0) | 5 (1.21) | 2 (0.47) |

**Table 3.** *Cont.*

| Keywords | I-PAAE, 2002I Andalusian Plan for Ecological Agriculture | II-PAAE, 2007II Andalusian Plan for Ecological Agriculture | III-PAPE, 2016III Andalusian Plan for Ecological Production |
|---|---|---|---|
| Control | 11 (1.98) | 6 (1.46) | 7 (1.63) |
| Cooperation | 1 (0.18) | 1 (0.24) | 2 (0.47) |
| Cooperatives | 3 (0.54) | 0 (0) | 2 (0.47) |
| Coordination | 3 (0.54) | 1 (0.24) | 2 (0.47) |
| Cost(s) | 3 (0.54) | 2 (0.48) | 1 (0.23) |
| Growth | 11 (1.98) | 7 (1.70) | 6 (1.40) |
| Development | 32 (5.75) | 35 (8.49) | 29 (6.76) |
| Local development | 0 (0) | 0 (0) | 1 (0.23) |
| Rural development | 1 (0.18) | 3 (0.73) | 6 (1.40) |
| Sustainable development | 0 (0) | 0 (0) | 1 (0.23) |
| Unemployed | 0 (0) | 0 (0) | 0 (0) |
| Gender inequalities | 0 (0) | 1 (0.24) | 0 (0) |
| Diffusion | 4 (0.72) | 2 (0.48) | 5 (1.16) |
| Distribution | 4 (0.72) | 1 (0.24) | 7 (1.63) |
| Diversity | 3 (0.54) | 0 (0) | 1 (0.23) |
| Dissemination | 2 (0.36) | 0 (0) | 1 (0.23) |
| Education | 0 (0) | 4 (0.97) | 0 (0) |
| Employment | 0 (0) | 4 (0.97) | 4 (0.93) |
| Company(ies) | 4 (0.72) | 3 (0.73) | 2 (0.47) |
| Strategy(ies) | 0 (0) | 0 (0) | 3 (0.70) |
| Local development strategies | 0 (0) | 0 (0) | 1 (0.23) |
| Europe | 3 (0.54) | 1 (0.24) | 0 (0) |
| Experience(s) | 9 (1.62) | 0 (0) | 1 (0.23) |
| Training | 31 (5.57) | 11 (2.67) | 13 (3.03) |
| Gender | 0 (0) | 8 (1.94) | 5 (1.16) |
| Management | 1 (0.18) | 2 (0.48) | 0 (0) |
| Large retail | 1 (0.18) | 0 (0) | 2 (0.47) |
| Rural Development Groups | 0 (0) | 0 (0) | 2 (0.47) |
| Gender equality | 0 (0) | 2 (0.48) | 3 (0.70) |
| Environmental impact | 0 (0) | 0 (0) | 0 (0) |
| Social impact | 0 (0) | 0 (0) | 0 (0) |
| Incentive(s) | 0 (0) | 0 (0) | 11 (2.56) |
| Information | 11 (1.98) | 2 (0.48) | 6 (1.40) |
| Innovation | 2 (0.36) | 0 (0) | 4 (0.93) |
| Research | 27 (4.86) | 8 (1.94) | 10 (2.33) |
| Young people | 0 (0) | 1 (0.24) | 3 (0.70) |
| Environment | 5 (0.90) | 2 (0.48) | 2 (0.47) |
| Rural environment | 1 (0.18) | 2 (0.48) | 0 (0) |
| Domestic market | 3 (0.54) | 2 (0.48) | 1 (0.23) |
| Internal market | 2 (0.36) | 6 (1.46) | 0 (0) |
| Market(s) | 31 (5.57) | 10 (2.43) | 7 (1.63) |
| Woman/Women | 0 (0) | 11 (2.67) | 10 (.33) |
| Norms | 9 (1.62) | 1 (0.24) | 0 (0) |
| Regulations | 7 (1.26) | 2 (0.48) | 4 (0.93) |
| Territorial planning | 1 (0.18) | 0 (0) | 1 (0.23) |
| Organizations | 6 (1.08) | 2 (0.48) | 5 (1.16) |
| Common Agricultural Policy | 3 (0.54) | 0 (0) | 0 (0) |
| Participation | 4 (0.72) | 9 (2.18) | 1 (0.23) |
| Participation of women | 0 (0) | 8 (1.94) | 1 (0.23) |
| Planning | 4 (0.72) | 4 (0.97) | 6 (1.40) |
| Policy(ies) | 6 (1.08) | 3 (0.73) | 2 (0.47) |
| Price(s) | 4 (0.72) | 1 (0.24) | 4 (0.93) |
| Organic production | 33 (5.93) | 45 (10.92) | 74 (17.25) |
| Productivity | 5 (0.90) | 0 (0) | 3 (0.70) |
| Producers | 10 (1.80) | 19 (4.61) | 1 (0.23) |

**Table 3.** *Cont.*

| Keywords | I-PAAE, 2002I Andalusian Plan for Ecological Agriculture | II-PAAE, 2007II Andalusian Plan for Ecological Agriculture | III-PAPE, 2016III Andalusian Plan for Ecological Production |
|---|---|---|---|
| Organic products | 30 (5.39) | 6 (1.46) | 20 (4.66) |
| Professional(s) | 4 (0.72) | 1 (0.24) | 4 (0.93) |
| Promotion | 7 (1.26) | 4 (0.97) | 5 (1.16) |
| Protection | 1 (0.18) | 6 (1.46) | 3 (0.70) |
| Natural resources | 0 (0) | 3 (0.73) | 0 (0) |
| Network(s) | 2 (0.36) | 0 (0) | 1 (0.23) |
| Regulation | 1 (0.18) | 0 (0) | 3 (0.70) |
| Profitability | 3 (0.54) | 1 (0.24) | 1 (0.23) |
| Residue(s) | 4 (0.72) | 0 (0) | 2 (0.47) |
| Environmental protection | 2 (0.36) | 0 (0) | 0 (0) |
| Responsibility(ies) | 1 (0.18) | 0 (0) | 0 (0) |
| Health | 0 (0) | 5 (1.21) | 0 (0) |
| Food safety | 4 (0.72) | 0 (0) | 0 (0) |
| Control system | 3 (0.54) | 2 (0.48) | 3 (0.70) |
| Certification schemes | 3 (0.54) | 0 (0) | 0 (0) |
| Sustainability | 0 (0) | 1 (0.24) | 1 (0.23) |
| Subsidy(ies) | 0 (0) | 0 (0) | 0 (0) |
| Maintainability | 0 (0) | 3 (0.73) | 0 (0) |
| Technique(s) | 9 (1.62) | 0 (0) | 3 (0.70) |
| Technology(ies) | 5 (0.90) | 6 (1.46) | 0 (0) |
| Territory(ies) | 2 (0.36) | 1 (0.24) | 3 (0.70) |
| Transfer | 5 (0.90) | 4 (0.97) | 3 (0.70) |
| Transformation | 12 (2.16) | 7 (1.70) | 4 (0.93) |
| Transparency | 0 (0) | 7 (1.70) | 4 (0.93) |
| Subtotal (words per column) | 556 (100%) | 412 (100%) | 429 (100%) |

Source: compiled by author. **Note:** The absolute frequencies were counted based on the existing words in Andalusian plans for organic farming in their sections: SWOT analysis (there is only one section in this regard in the first and third plan), objectives, and actions and/or interventions (present in the three plans). Those sections are considered essential to identify the values present in the three Andalusian plans for organic farming. When it was relevant, the search was conducted by adding the appearance of the Spanish word in the singular and in the plural forms; for example, "water" and "waters" added up to 2 reiterations in the I Andalusian Plan for Ecological Agriculture [6]; however, these same words did not appear at any time in the III Andalusian Plan for Ecological Production [8]. For their part, the relative frequencies are expressed in percentages and in parentheses and were calculated based on the total number of words in each column, that is, on the sum of the repetitions of the keywords selected in the sections of each plan analyzed. The percentages were rounded to the second decimal.

## 3. Methods

The main methodology used in this article is content analysis, the method "most frequently used in many social sciences, acquiring an unprecedented significance as computerized procedures were introduced in data treatment" [51] (p. 2). The basic purpose of content analysis is to systematically study the "manifest and later latent [content] applied to different topics and themes" [52] (p. 129). Within content analysis, various types of analysis are differentiated [53,54]. Specifically, the so-called thematic content analysis,

> "*considers the presence of terms or concepts, regardless of the relationships that have emerged between them. The most widely used techniques are frequency lists, thematic identification and classification, and word search in context. Perhaps the most frequent is to search—and eventually analyze more carefully, with another technique—units in which a certain theme appears*" [53] (p. 20).

In this way, through the quantitative thematic content analysis [51–54] elaborated in this article, an attempt was made to identify the main themes and values that appear in the three Andalusian plans for organic farming [6–8], both in the full texts of the plans (Table 1) as well as in their sections on objectives, measures, actions, strategic lines, and SWOT (strengths, weaknesses, opportunities, and threats) analysis (Table 3). To this end, two tables were drawn up in which the topics covered in the aforementioned plans are shown and the times that such words are repeated are quantified using the keywords selected in said

tables, in absolute and relative frequencies. In the search for keywords, the computer tool to search for words in PDF documents ("edition," "search") of the three analyzed plans was used. Table A1 (see Appendix A) shows the keywords used for the content analysis of the three plans: These words were codified in Spanish and English. All this implied artisan and systematic work facilitated by said tool and Table A1. In addition, several versions of Tables 1 and 3 were prepared prior to the ones presented here, which allowed the exhaustive identification of keywords and their analysis.

The first phase of the analysis presented graphically how the appearance of the various themes (keywords) evolved quantitatively in the three complete documents studied (see Table 1). Furthermore, as the Andalusian plans for organic farming are official administrative texts, they supposedly emanated from certain negotiations and agreements between the different sociopolitical agents interested in organic farming, officially represented in the Consejo Andaluz de la Producción Ecológica (Andalusian Council of Ecological Production, CAPE). Thus, the themes appear structured in the objectives, measures, actions, and strategic lines of such plans (see Table 3). This structure was relevant since it follows a set of interests, preferences, and social values of said agents. That is to say, the themes appear hierarchically and ordered according to the explicit and agreed-upon priorities, to a greater or lesser degree, by certain subjects in a changing socio-historical scenario, such as the Andalusian ecological agriculture of the past decades.

Likewise, as Tables 1 and 3 show the evolution of these priorities, which can be treated as implicit or explicit social values around ecological agriculture, it was also possible to link such priorities to the discursive social context in which the themes appear. Thus, different discursive contexts can legitimize the appearance of certain values, represented in keywords, to the detriment of others. In other words, quantitative thematic content analysis allows certain discursive social frameworks to be inferred that legitimize and/or justify the appearance of the themes in the analyzed texts.

A qualitative semantic content analysis was carried out in the second phase of the analysis [53] (p. 21) to identify the structure of established meanings linking the themes (see Table 2). Therefore, "all the occurrences [absolute and relative frequencies] that agree with said structure are repeatedly and systematically studied. Semantic analysis aims to study the relationships between topics covered in a text. For this, the relationship patterns that will be considered must be defined" [53] (p. 21).

The relationship patterns considered in this article are, on the one hand, causally linked to the theoretical perspectives, approaches, and values on organic farming set out in the previous section, and on the other, illustrate the evolution of Andalusian organic farming in the past decades. This second phase of analysis allowed for two tasks: first, the identification and relationship of the exposed themes in the analyzed documents and second, the linking of the themes to various approaches to organic farming. This link was made by categorizing the keywords selected in the three documents analyzed, and was due to the appearance, or not, of such keywords. That is to say, the empirical–analytical logic of the greater or lesser presence and/or absence of said keywords was followed, and such approaches were named accordingly. In this way, on the one hand, two general approaches to ecological agriculture emerged, namely, an ecologist approach and a productivist–technocratic approach. On the other hand, several semantic fields related to the general approaches were identified, which, in turn, showed the increasing or decreasing presence and omission of the keywords in the three plans analyzed (Table 2). The Andalusian administration incorporated these two approaches to different degrees, or was even able to marginalize some of them, in Andalusian plans for organic farming. This links to the social and political context and, therefore, to the predominant or hegemonic social discursive frameworks in the times the plans were published (2002, 2007, and 2016). Thus, this article contributes to the understanding of trends in the political planning of Andalusian ecological agriculture during the past decades.

Indeed, as the goal was to understand the values linked to ecological agriculture in Andalusian plans for organic farming, it was also necessary to understand the specific

meanings that such values acquire, since they marked the political priorities of the agents involved in the elaboration of the plans. Such meanings, present in the identified semantic fields, probably changed as the socio-historical context and the social position of the aforementioned agents was transformed [55–60]. In other words, socio-political subjects can strategically redefine their preferences or priorities regarding ecological agriculture over time as their socio-economic interests and positions of power change [9,61,62]. That is, social actors tend to modify their social priorities and values if their social position changes in the contexts in which they live; this position is also conditioned by the institutional and regulatory framework in which the activity carried out takes place [10]. Thus, the qualitative semantic content analysis of the plans revealed certain webs of meanings produced by "interested" and "positioned" subjects in Andalusian ecological agriculture. This was a first step in grasping the social discourses addressing ecological agriculture.

The methodology applied in scientific research may have certain advantages and disadvantages. Specifically, the advantages of the content analysis elaborated in this article were the following: (1) the contents of the Andalusian plans for organic farming had not been systematically investigated before. (2) This study facilitates a detailed understanding of the main themes covered in the plans. (3) The classification of the themes was based on theoretical criteria and empirical knowledge. (4) Various approaches and values on ecological agriculture were detected in these plans. (5) Six semantic fields have been established that allow the socio-political themes and values shown in each semantic field to be compared [51–54]. Among the disadvantages of content analysis, the following were considered: (1) the limitation of the number of keywords analyzed in the form of nouns: 109 nouns were selected according to theoretical sampling. (2) The specific linguistic context in which the keywords appeared was not considered, since this implies much broader research than the one presented here. (3) Keywords in the form of verbs were not analyzed to identify the themes of the plans studied. (4) It is likely that a content analysis based on keywords in the form of verbs and nouns could indicate other themes not considered in this article (see Section 5.4). (5) Despite the limitation of the keyword sample, the presentation of the data was generated in large tables.

Lastly, it became clear that in the I-PAAE [6] (pp. 69–78) and in the III-PAPE [8] (pp. 32–36) a SWOT analysis of Andalusian ecological agriculture was carried out. SWOT (strengths, weaknesses, opportunities, and threats) analysis is an analytical tool that aims to make visible the issues or themes considered a priority for the individuals and human groups involved in an economic activity and, therefore, the strategies and expectations can also be reflected in the analysis, including the values attributed to the economic activity [63–65]. In short, the approaches to ecological agriculture, which the individuals implicitly or explicitly show, are different and opposed to the extent that they prioritize some socio-political values to the detriment of others.

## 4. Results

### 4.1. Contextualization of Andalusian Organic Agriculture

The number of hectares dedicated to organic farming in Europe has grown steadily, "mainly as a result of the institutional recognition that the sector had in the mid-1990s" [34] (p. 247). According to Boza, "one of the factors that has most motivated this growth has been the strong adherence to organic farming on land managed by producers from Central and Eastern European countries that became part of the European Union in May 2004" [34] (pp. 247–248). This same phenomenon is found in Spain and, very particularly, in Andalusia, the Spanish region with the largest extension devoted to this agricultural activity. In addition, the traditional importance that the agricultural sector has played in Andalusian society means that the Common Agricultural Policy (CAP) "has been one of the aspects of community management of greatest interest to the region since the accession of Spain as a member state" [35] (p. 301) of the European Economic Community (EEC), in 1986, now the European Union (EU).

In this way, Andalusia has been the beneficiary of significant European public aid to develop and modernize regional agriculture. The promotion and development of Andalusian ecological agriculture has benefited, since the 1990s, from the institutional framework and public policies undertaken by the EU, managed in Spain by the autonomous governments. In that decade, the Common Agricultural Policy (CAP), until then based on a system of direct aid to production, underwent important changes due to the high agricultural surpluses and the pressure from the World Trade Organization to promote competitive imports [35,41,66]. Such changes were aimed at supporting an extensive production model, more diversified and linked "with the protection of the environment and with the promotion of rural development. However, each member state has arranged the specific strategies within this framework of action in a discretionary manner" [35] (p. 294).

Likewise, the growth of organic farming in Andalusia has been conditioned by the rural development strategy and by European agricultural public policy, specifically by the CAP [35,41]. In this sense, the main axes of the Andalusian Rural Development Program (2007–2013) were three: increasing agricultural competitiveness, increasing the quality of life in rural areas, and improving the environment [67]. The program states that aid for organic farming is consistent with the objectives of the European Agricultural Fund for Rural Development (EAFRD). Thus, the Rural Development Program is clearly in tune with public support for Andalusian organic farming, which is involved in the specific plans for the sector [35] (pp. 301–302). These plans were devised and promoted by the Junta de Andalucía in three successive stages during the first two decades of this century: 2002–2006, 2007–2013, and 2014–2020. The stages correspond to the three Andalusian plans for organic farming considered in this study [6–8]. The economic resources mobilized by these plans have been essentially concentrated in the supply or in the producer sector—support for organic production—which has made difficult "the construction of a sector more independent of public aid" [34] (p. 163). In fact, the three plans have dedicated 70% or more of their respective budgets "to supporting organic production through agri-environmental aid" [35] (p. 302). Specifically, the III-PAPE allocated 78% of its budget to incentives for organic production [8] (p. 107). This reproduced the productivist, economistic, and technocratic orientation of Andalusian public policies on ecological agriculture.

Therefore, the regional administration (Junta de Andalucía), responsible for planning organic farming in Andalusia between 2002 and 2016, has emphasized and prioritized the themes and/or issues that have directed its public policies on ecological agriculture. Accordingly, the three plans studied include data, tables, and graphs on the economic resources destined to the promotion of regional organic production. As stated in Section 3, in two of the three plans studied—the first and third plans—SWOT (strengths, weaknesses, opportunities, and threats) analysis was included that allowed said administration to classify and prioritize relevant themes for Andalusian ecological agriculture. As mentioned above, the themes presented in the plans are highly conditioned by the rural development strategy and by European agricultural public policy, specifically by the CAP. For these reasons, the following keywords were included in the empirical analysis: "development," "rural development," "Common Agricultural Policy," "Europe," "policy(ies)," "strategy(ies)," "local development strategy," etc. Other keywords were also included in the content analysis: "water," "grant(s)," "organic farming," "agroindustry(ies)," "food," "quality," "competitiveness," "gender," "Rural Development Groups," "incentive(s)," "innovation," "environment," "rural environment," "woman/women," "participation," "organic production," "producers," "food safety," etc. These words had a relatively important presence in the Andalusian Rural Development Program (2007–2013) [67]. In short, all the keywords included in the empirical analysis were inferred, on the one hand, from the data and texts presented in the three plans analyzed, and on the other, from the practice and evolution of ecological agriculture in Andalusia during the first two decades of this century. That is to say, the selected keywords were significant and relevant to understanding in greater detail the themes considered by the plans studied.

For its part, the Andalusian Rural Development Program 2014–2020 categorized this community as a "transition region," according to article 2 of the European Commission Implementing Decision of 18 February 2014, which set out the "list of regions eligible for funding from the European Regional Development Fund and the European Social Fund and of Member States eligible for funding from the Cohesion Fund for the period 2014–2020" [68] (p. 11). Furthermore, this program conceived of organic production as a key area of rural development and a priority criterion in most investment measures.

In addition, it is significant that the Andalusian Plan for Ecological Agriculture (I-PAAE) was "the first document of this nature" prepared in Spain at the beginning of this century [6] (p. 6). According to what was stated in the I-PAAE, this plan was original in its conception and also in the way it was carried out: "through dialogue, the contribution and the consensus of all the economic and social agents involved in the ecological agriculture sector" [6] (p. 6). For this reason, this first public document on organic farming in Andalusia, together with the two following plans for organic farming, needed to be analyzed in detail from a perspective that considers the underlying socio-political values in these official texts, since they show the approaches to, conceptions of, and ideas about ecological agriculture.

The development of organic farming in Andalusia is supported by multiple regional regulations that try to adapt European and Spanish standards on said agricultural activity to the Andalusian territory. The different systems of control and certification of organic production developed in Andalusian are relevant in this regard. Thus, through Royal Decree 1852/1993 of 22 October, the decentralization of the control of organic farming in Spain was approved, and Andalusia was the first autonomous community to assume this competence [69]. At the same time, through the Order of 23 November, 1993, "the regional Ministry of Agriculture and Fisheries recognizes the Andalusian Territorial Committee for Ecological Agriculture as the only control body for organic farming in Andalusia. Shortly after, through the Order of 9 August, 1994, the Territorial Committee was renamed the Comité Andaluz de Agricultura Ecológica (Andalusian Committee for Ecological Agriculture, CAAE)" [34] (p. 144). Among the functions assigned in the mid-1990sto the CAAE were the promotion and control of Andalusian organic farming, the organization of training activities among farmers, and the preparation of studies and other activities designed to expand the organic agricultural market.

As the number of operators linked to organic farming increased in Andalusia, the CAAE asked the regional Ministry of Agriculture and Fisheries "to start acting as a private control body under the name of the Asociación Comité Andaluz de Agricultura Ecológica (Andalusian Committee for Ecological Agriculture Association)" [34] (p. 151). Its effective authorization was made public on 23 April 2003. In the practice of the certification of organic agriculture, the CAAE Association has been the most important in Spain, since "it is the control authority with the largest number of subscribed operators, as well as surface. In 2008, 714,663 hectares dedicated to organic farming were registered in the CAAE Association, seven times more than in 2001, as well as 6813 operators" [34] (p. 152). This association has acquired an international dimension, being the entity that certifies the largest organic production area in Europe (more than 1 million hectares), expanding its scope of action to other Spanish autonomous communities, such as Castilla-La Mancha [70]. Additionally, there have been in Andalusia, since the end of the 1990s, other private agencies for the control and certification of organic farming, such as Sohiscert, Agrocolor, Agrocalidad del Sur, etc.

For its part, according to article 4 of Decree 166/2003 of 17 June on organic agri-food production in Andalusia, the "Andalusian Council of Ecological Production is created, as a collegiate and consultative body in the elaboration of standards and in the setting of criteria for the application, in the territory of the Autonomous Community of Andalusia, of the provisions on organic production" [71]. According to said decree, the Consejo Andaluz de la Producción Ecológica (Andalusian Council of Ecological Production, CAPE) is to meet every six months and develop "advisory functions in the matter of organic agri-food production, mainly in the monitoring of strategic plans, in the preparation of reports

on community agricultural policy, in promotional campaigns and in any other activities related to organic agri-food production deemed necessary" [71]. In other words, CAPE is an Andalusian institutional entity that aims to regulate the development and operation of regional ecological agri-food production [72]. For this reason, the agreements reached in this council define the regulatory framework of the production processes developed in Andalusian organic farming. Thus, the decision-making process in the elaboration of the aforementioned regulatory framework and organic farming plans is of a neo-corporate type; that is, they are carried out in a concerted manner between the public administration (Regional Ministry of Agriculture and Fisheries, Junta de Andalucía) and a select group of agents, such as socio-economic organizations, unions, political officials, public officials, etc. [73,74].

Indeed, the CAPE is made up of various actors with influence on the regulation of Andalusian agriculture: "(a) A President. This position will be held by the head of the Consejería (Regional Ministry) of Agriculture and Fisheries. (b) A Vice President. This position will be held by the head of the General Secretariat of Agriculture and Livestock of the regional Ministry of Agriculture and Fisheries" [71] (article 4, p. 13.604). Likewise, in accordance with Decree 242/2003 of July 29, correcting the errors of Decree 166/2003, [75], the CAPE is to be made up of numerous members representing the main interests of regional ecological agriculture: the Andalusian autonomous administration (the head of the Directorate General of Industries and Food Promotion; two officials from the Ministry of Agriculture and Fisheries, appointed by the head of the same; a representative of the Ministry of the Interior; two officials from the Ministry of the Environment; and three representatives of the control bodies with the highest volume of activity among those authorized by the Ministry of Agriculture and Fisheries); two members elected by the most representative union organizations; two representatives of the agri-food industry appointed by the Andalusian Businessmen Confederation; two representatives of the Andalusian Consumer Organizations; several agents of the Andalusian agriculture business community (Andalusian Federation of Agricultural Cooperative Companies, Federation of Agricultural Associations-Young Farmers of Andalusia, Coordinator of Organizations of Farmers and Ranchers of Andalusia, Union of Small Farmers of Andalusia); two representatives of the universities and research centers or people of recognized standing from the organic production sector; as well as an official from the Directorate General of Industries and Food Promotion, with the category of Head of Service, who is to act as secretary.

Furthermore, by virtue of Law 1/2003 of 10 April, the Instituto Andaluz de Investigación y Formación Agraria, Pesquera, Alimentaria y de la Producción Ecológica (Andalusian Institute for Research and Training in Agriculture, Fisheries, Food and Ecological Production) was created. This body responds to the relevance that "the agricultural, fishing and food sectors have for Andalusian society and economy, as well as the need for their modernization to preserve and expand their social and economic projection" [76] (p. 9.324). Among the main functions of that body are the following: "(a) Support the development of policies for agriculture, fisheries, food and organic production of the Junta de Andalucía in the scientific and educational fields. (b) Design and carry out sector research plans, with the participation of the agents involved, taking into account the objectives, programs and instruments of the Technological Research and Development Plans in force at all times in Andalusia. (c) Plan and carry out information and training programs for farmers, fishermen, workers and technicians through technology transfer, based on the results of own or third-party research or other sources of knowledge, as well as evaluating their results depending on the degree of adaptation of those technologies" [76] (p. 9.325).

Finally, in 2004, the Dirección General de Agricultura Ecológica (Directorate General of Ecological Agriculture, DGAE) was created as a dependent body of the Junta de Andalucía. Manuel González de Molina, university professor expert in agroecology [77], "recognized environmental activist and member of the Verdes de Andalucía (Andalusian Green Party), is appointed director of the institution thanks to an agreement between said political force and that ruling in the Community" [34] (p. 221). At the beginning of its activity,

the aforementioned Directorate sought to identify the demands of the agents involved in Andalusian ecological agriculture [34] (p. 221). To that end, numerous meetings were organized in Andalusia and different farmers from the mountain range areas of Serranía de Ronda and Sierra de Segura, who later joined the Participatory Guarantee Systems project, posited against "the need to apply the official certification system in their case, demanding a possible solution from the Directorate, in order to make their agricultural activities viable" [34] (p. 221). In this context, the aforementioned Participatory Guarantee Systems (SPG) become relevant, since organic farmers can intervene in the certification processes of organic production incorporating their views and interests. However, at the end of 2007, the aforesaid professor "ceases to be director of the DGAE and the SPG project in Andalusia loses a large part of its institutional support" [34] (p. 229).

*4.2. Analysis of the Approaches, Themes, and Values in the Planning of Andalusian Ecological Agriculture*

Table 1 shows the evolution of the main themes present in the three Andalusian plans for organic farming. A total of 109 keywords or nouns that appear in said plans were selected and codified, by means of which an attempt was made to identify the current socio-political approaches and values in the political planning of Andalusian ecological agriculture (see Table A1 in Appendix A). Next, the content analysis of these themes structured in various semantic fields is presented. These are not totally opposed to each other, but are complementary: the treatment of the themes, approaches, and explicit values of the plans studied revolves around these semantic fields, as shown in Table 2.

4.2.1. The Ecologist Approach in Organic Farming Plans

The central axis of the first semantic field is the agroecological approach: This, in a classificatory attempt, was called semantic field A or the illegitimate ecologist approach (see Table 2). This semantic field decreased its presence and became a minority, especially when moving from the first to the third plan studied, depending on the number of times its most representative words appeared repeated. In semantic field A words such as "water," "agroecology," "organic farming," "diversity," "environmental impact," "social impact," "environmental protection," and "food safety" appeared. In this regard, the few mentions of a fundamental natural resource such as "water" were very revealing: seven references in the I-PAAE and no reference to this same term in the III-PAPE; it seems as if water is an unquestionable, or unlimited, resource in an area such as Andalusia, traditionally characterized by droughts and water scarcity. This raised a contradiction. Furthermore, significant words from semantic field A, such as "organic farming" and "agroecology," went from 681 (18.65%) and 11 (0.3%) reiterations in the I-PAAE to only 26 (1.06%) and 3 repetitions (0.12%) in the III-PAPE. The decrease in the presence of both words was very remarkable. For its part, it was striking that three basic words for ecological agriculture such as "diversity," "environmental protection," and "food safety" had such a low presence in the III-PAPE, appearing only once each. Finally, it was very relevant that said plan (III-PAPE) did not refer to the environmental impact and/or social impact, when all agricultural activity has an effect on the community where it is practiced. This indicates that such terms became real taboos, so that some of them were not named in the latest organic production plan.

On the other hand, semantic field B, here called the legitimate ecologist approach, was made up of themes present in agroecology, but on this occasion was composed of words that, although not the majority, acquired gradually more presence or visibility as we moved from the first to third plan of organic farming. Thus, such words seemed to be incorporated into the discursive scope of the media agenda and politically correct language [78] and, therefore, the Andalusian administration has tried to give them visibility in the plans analyzed. Semantic field B was made up of the following words included within the aforementioned politically correct language: "organic food," "animal welfare," "biodiversity," "short supply chains," "climate change," "citizenry," "traditional knowledge," "conservation," "local development," "sustainable development," "gender inequalities,"

"local development strategies," "gender," "Rural Development Groups," "gender equality," "woman/women," "participation of women," "natural resources," "sustainability," and "maintainability." All of them are words that increased their appearance between the I-PAAE and the III-PAPE. This was especially significant in the cases of words that did not appear in the first organic farming plan and appeared with high frequency in the third plan. This was the case with "biodiversity," "short supply chains," "climate change," "citizenry," "local development," "local development strategies," "gender," "gender equality," and "participation of women." These last three words, linked to the presence that women acquired in the II-PAAE as one of its strategic axes, made a relevant appearance in the third plan. Regarding the words "traditional knowledge," "local development," and "sustainable development," which were either not mentioned in the first two plans ("traditional knowledge" and "local development") or were only mentioned once ("sustainable development"), in the III-PAPE the first word appeared twice (0.08%), the second was displayed 13 times (0.53%), and the third word was repeated 12 times (0.49%). In this way, the III-PAPE was characterized as the Andalusian organic production plan committed to "sustainability," so this word went from appearing only four times (0.11%) in the I-PAAE to being reiterated 15 times (0.61%) in the III-PAPE. This change is significant because it implies the incorporation of new social values in the last Andalusian organic production plan: These values include sustainable and local development, the participation of women, and the commitment to gender equality and the sustainability of ecological agriculture.

Likewise, a third semantic field was observed (field C), called a neutral ecologist approach, in which the terms that made it up did not show a clear evolution of increase or decrease in their appearance, but rather maintained an irregular variation. In this semantic field there were words such as "pollution," "environment," "domestic market," "internal market," "protection," "participation," and "health." All of them were socially relevant words that were taken for granted, in the development of ecological agriculture, but that did not undergo significant changes in the three plans analyzed. This may be due to the fact that the administration assumed that such terms were associated with central aspects of said agrarian practice. However, the terms "pollution," "health," and "internal market" had a clearly greater appearance in the II-PAAE than in the other two plans. This may be due to the fact that the second plan proposed ambitious goals and changes that were not achieved later [21,49].

4.2.2. The Productivist Approach in Organic Farming Plans

This approach to ecological agriculture was predominant in quantitative terms, especially as we moved in the analysis from the first to the third plan. As previously mentioned, the I-PAAE clearly supported this approach to ecological agriculture and within it, three semantic fields that appeared in the three Andalusian plans for organic farming could be distinguished.

On the one hand is semantic field D, which was made up of themes related to the administration and/or management of organic farming that grew, in absolute and relative frequencies, as we moved from the first to the third Andalusian organic farming plan. In the semantic field D or legitimate productivist–technocratic approach, the following terms were shown, which were those that enhanced the prevailing ecological agriculture approach chosen by the Andalusian administration, present in the three plans and especially in the third: "agroindustry(ies)," "counseling," "benefit(s)," "competitiveness," "commercialization," "development," "rural development," "employment," "company(ies)," "strategy(ies)," "management," "incentive(s)," "information," "innovation," "young people," "rural environment," "regulations," "planning," "organic production," "professional(s)," "territory(ies)," and "transparency." These themes represented the vision and values of ecological agriculture prioritized by the regional administration and, therefore, was a vision that intended to transfer and impose itself on the group of agents involved in Andalusian ecological agriculture. It should be noted, in the case of words such as "counseling," "competitiveness," "development," "rural development," "innovation," and "transparency,"

that the percentages of their appearance in the I-PAAE were, respectively, 0.33%, 0.05%, 3.67%, 0.33%, 0.08%, and 0.03%, whereas in the III-PAPE, such percentages became 1.43%, 0.86%, 7.52%, 1.76%, 0.69%, and 0.53%, respectively. Undoubtedly, the central social values of this approach to ecological agriculture were competitiveness, rural development, innovation, and transparency. For its part, the word "incentive(s)" went from one mention (0.03%) in the first plan to being repeated 63 times (2.57%) in the third plan. Furthermore, the term that appeared most frequently in the last plan was "organic production": repeated 138 times in the first plan (3.78%) and with 396 mentions in the third plan (16.18%), so that whereas the I-PAAE referred to "organic farming," the III-PAPE referred above all to "organic production." Ecological agriculture broadened its meanings and extended beyond the agrarian–peasant practice. This emphasizes the economic and productivist orientation that the autonomous administration granted to ecological agriculture, without considering the sectoral and regional imbalances of said agrarian practice. The administration conceived of ecological agriculture as another economic activity that generates employment for groups with high unemployment rates, such as women and young people in rural areas, and a sector that yields quality products differentiated from conventional agriculture, although it has the same dependencies as the latter [49,66].

On the other hand, in semantic field E, called the illegitimate productivist–technocratic approach, a set of words related to social and political values that were in discursive decline were shown. A different issue was whether this had a real translation in the public policies developed for Andalusian organic farming. Semantic field E was structured around the following terms, which had a great echo in the first plan and which seemed to have fallen into a certain political–institutional disgrace in recent times: "public aid," "beneficiaries," "concentration," "supply concentration," "consumer(s)," "cooperatives," "cost(s)," "growth," "Europe," "training," "large retail," "research," "market(s)," "norms," "Common Agricultural Policy (CAP)," "price(s)," "producers," "profitability," "subsidy(ies)," and "transformation." This was the case with the terms "grant(s)," "public aid," "growth," and "subsidy(ies)," which made a clear appearance in the I-PAAE (174, 4, 56, and 12 reiterations, respectively) and were nevertheless avoided, in comparative terms, in the III-PAPE (29, 1, 16, and 2 repetitions, respectively). In the last Andalusian organic production plan, instead of "grant" for ecological agriculture, which was and remains highly dependent on "public aid," it seems to be more legitimate to speak of "incentive(s)" for production, as stated above. It is also very significant that the word "Europe," from which much public aid for the Spanish agricultural sector has come through the CAP, only appeared 3 times (0.12%) in the third plan, whereas it was repeated 49 times (1.34%) in the first plan. This shows the role recently played by public subsidies in ecological agriculture. Perhaps due to the budget cuts derived from the economic crisis of 2008, speaking of public aid and subsidies is not regarded as legitimate.

Finally, a sixth semantic field (F) was identified as linked to the ecological agriculture approach chosen by the Andalusian administration: the neutral productivist–technocratic approach in which certain key words such as "administration(s)," "control," "organizations," and "policy(ies)" were naturalized. This implied an attempt to avoid conflict in spheres that precisely regulate social conflicts and the struggle for values. The words of this semantic field did not present a clear variation; either they remained stable or with little relevant or irregular variations between the three plans. In this semantic field, different words that the autonomous administration incorporated in an uncritical way and without questioning were identified, since they constituted the "cultural arbitrary" [79] of the management of ecological agriculture and, therefore, formed the central nucleus of the administrative management routine. These words were "administration(s)," "food," "quality," "certification," "knowledge," "control," "consumption," "communication," "coordination," "cooperation," "unemployed," "distribution," "diffusion," "dissemination," "education," "experience(s)," "organizations," "policy(ies)," "promotion," "network(s)," "regulation," "residues," "responsibility(ies)," "control system," "technique(s)," "technology(ies)," and "transfer." The emerging social values in this semantic field were quality,

control, coordination, cooperation, and regulation, because the practice of organic farming requires a rigorous and internationally recognized system of control and certification of the quality of organic products. In fact, a large part of these products are destined for export, due to low demand in the Spanish and Andalusian markets [35].

### 4.2.3. The Ecologist Approach in the Main Sections of Organic Farming Plans

Table 3 shows to what extent the previous semantic fields were present or not in the central sections of the Andalusian plans for organic farming. Thus, the study focused exclusively on content analysis in the following sections of the aforementioned plans: objectives, actions, and/or interventions, and SWOT analysis. The latter only appeared in the first and third plans, whereas objectives, actions, and/or interventions were present in all three plans.

Regarding the evolution of semantic field A in Table 3, the trends mentioned above were confirmed and intensified, to the point that the so-called illegitimate ecologist approach was excluded or disappeared in the sections of objectives, actions, and/or SWOT analysis of Andalusian plans for organic farming. This was the case with words such as "agroecology," "social impact," and "environmental impact." Other words such as "water," "environmental protection," and "food safety" disappeared in the II-PAAE and the III-PAPE. It is paradoxical that words such as "water," "social impact," and "environmental protection" disappeared from the last two plans, whereas other words of the semantic field had a marginal appearance, as was the case of "organic farming," which was only mentioned three times in III-PAPE.

The trend followed in semantic field B or the legitimate ecologist approach was partly reaffirmed by the data in Table 3. Thus, several words from said semantic field showed the following evolution: absence in the I-PAAE and allusion and/or reiteration in the III-PAPE. This was the case for the following words: "biodiversity," "short supply chains," "climate change," "citizenry," "local development," "sustainable development," "gender," "Rural Development Groups," "gender equality," "woman/women," "participation of women," and "sustainability." The new appearance of different words in the III-PAPE, which were not present in the I-PAAE, was especially significant in the cases of terms referring to the presence of women in ecological agriculture. This happened with "gender," "gender equality," "woman/women," and "participation of women." Therefore, it was confirmed that the values linked to the legitimate ecologist approach were sustainable and local development, gender equality, and the participation of women.

In reference to semantic field C or the neutral ecologist approach, according to Table 3, the following was observed: The words in this semantic field, in general, appeared more often in the II-PAAE, whereas the appearances of the same in the III-PAPE were null or not relevant. This was observed in the case of significant words in this semantic field such as "pollution," "internal market," "participation," "protection," and "health." Therefore, the values linked to semantic field C (protection, participation, and health) were in connection with the II-PAAE and, in part, detached from the other two plans, especially the I-PAAE. Thus, such values, especially health and protection, were not fundamental in the set of the three plans analyzed, since they appeared almost isolated in the II-PAAE.

### 4.2.4. The Productivist Approach in the Main Sections of Organic Farming Plans

In relation to semantic field D or the legitimate technocratic–productivist approach, Table 3 confirms that this was the approach to ecological agriculture prioritized by the Andalusian administration, and was clearly predominant in the third plan. Thus, words such as "agroindustry(ies)," "competitiveness," "commercialization," "rural development," "employment," "strategy(ies)," "incentive(s)," "innovation," "young people," "planning," "organic production," and "transparency" appeared with much greater reiteration in the III-PAPE, both in absolute and relative frequencies. For this reason, perhaps the values identified in semantic field D (competitiveness, rural development, innovation, and transparency) referred, above all, to the III-PAPE, which opted for a productivist and

technocratic vision of organic farming, excluding the agroecological approach and taking advantage of the great growth experienced by said agriculture, both in cultivated area and in producers, since the beginning of the 21st century.

The evolution of semantic field E, or the illegitimate productivist–technocratic approach, showed an evident discursive regression. This was the case, according to Table 3, with the words "grant(s)," "public aid," "concentration," "supply concentration," "cost(s)," "growth," "Europe," "research," "market(s)," "norms," "Common Agricultural Policy (CAP)," "producers," "profitability," "subsidy(ies)," and "transformation." That is to say, the values associated with semantic field E (public aid, growth, Europe, and profitability) seemed to be part of the past (I-PAAE) rather than the present (III-PAPE). Said productivist approach became illegitimate for the administration itself, as well as in the central sections of the plans studied.

Overall, taking into account Table 3, semantic field F showed an irregular evolution. Thus, on the one hand, a series of words from this semantic field were observed that seemed to be complementary to semantic field D and, therefore, made up themes and values assumed in the approach prioritized by the Andalusian administration in the three organic farming plans. These words, which showed a more stable evolution in the three plans and which were part of the values of semantic field F, were "administration(s)," "food," "control," "coordination," "organizations," "promotion," "control system," and "transfer." On the other hand, there were several words in this semantic field that experienced a growing trend between the first and the third plan, such as the word "consumption," which went from 10 repetitions in the I-PAAE to 20 repetitions in the III-PAPE, whereas words such as "cooperation," "distribution," "diffusion," and "regulation" showed slight increases between the first and third plans analyzed. Finally, another group of words from semantic field F adopted a decreasing evolution between the I-PAAE and the III-PAPE: This was the case for the terms "quality," "knowledge," "dissemination," "experience(s)," "policy(ies)," "residues," "responsibility(ies)," "technique(s)," and "technology(ies)."

## 5. Discussion and Conclusions

### 5.1. Main Findings

The study of approaches to, and of socio-political values pertaining to, ecological agriculture is relevant in the context of Andalusia, since a significant environmental movement and agrarian social thought have developed in this region in recent decades, with important consequences for the ideas and practice of organic farming [17,24,31,43,80–83]. In this regard, this investigation provides detailed knowledge of themes, approaches, and values developed in the three Andalusian plans for organic farming. The content analysis of these plans shows that the Andalusian administration, represented by the regional Ministry of Agriculture and Fisheries of the Junta de Andalucía, as well as the social, economic, and political agents that make up the Andalusian Council of Ecological Production (CAPE), have prioritized a productivist–technocratic approach to organic farming, to the detriment of a more holistic, sustainable, and local approach more attentive to the various interdependencies of rural communities, such as that symbolized by the illegitimate ecologist approach (semantic field A). Although this last approach is partly present in the three plans, especially in the first two, in the last plan a productivist approach to ecological agriculture is clearly preferred. In this way, the change in the denomination of the III Andalusian Plan for Ecological Production (III-PAPE) is very revealing, with a shift in the text from organic farming to emphasizing organic production. This seems to have important implications in the definition of the new socio-political values configured around ecological agriculture in the III-PAPE. Such values are linked to so-called and semantic fields D and F that make up the legitimate and neutral productivist-technocratic approaches.

These approaches, prioritized by the Andalusian administration and by the various actors of ecological agriculture, promote and highlight values such as competitiveness, rural development, innovation, transparency, quality, control, and regulation, to the detriment of agroecology, diversity, respect for the environment, and food safety. It is striking that in the

III-PAPE the words social impact, environmental impact and water were not mentioned, despite the fact that agricultural activity depends on this natural resource and generates local impacts of a different nature. Consequently, in the III-PAPE a conventionalized and rational-technical vision of the ecological rural environment was imposed, which conceived of ecological agriculture as a subsector of the Andalusian economy that offers a quality product and differentiated from conventional agriculture [49,50]. Thus, the practice of organic farming is subordinated to the same logic and interdependencies as conventional agriculture [34,35]. Therefore, the values of quality, control, coordination, cooperation, and regulation were chosen (semantic field F). All of them are functional values to develop and legitimize, by the Andalusian administration and by the actors participating in the elaboration of the plans, that conventionalized and instrumental vision of organic farming [82–85].

Indeed, hypotheses 1 and 2 were confirmed, since the context in which ecological agriculture developed in Andalusia, from the 1990s, has been conditioned by several changes that have favored this vision of organic farming in the region. First, this is the Spanish region that has accumulated the largest area dedicated to organic farming since the early 2000s. Second, this region pioneered the development and implementation of the first organic farming plan in Spain, published in 2002 [6]. Third, the growth of this agricultural activity was promoted by European public aid from the Common Agricultural Policy (CAP), which, in turn, stimulated the industrialization and modernization of regional agriculture. In this way, this growth was guided by the rural development strategy of the CAP, which directed its policies to promote competitiveness, as well as to improve the quality of life and the environment in Andalusia [67]. Fourth, the growth and modernization of Andalusian ecological agriculture were conditioned by multiple European norms and regulations that, in turn, promoted a clearly productivist, bureaucratized, and technocratic orientation of the public policies on ecological agriculture implemented in Andalusia [35,66]. In this sense, it is coherent that the aforesaid vision of ecological agriculture should prevail, as well as the approaches and values exposed in semantic fields D and F.

Thus, as an answer to the research question, the empirical analysis of the three plans shows a changing evolution in the themes, approaches, and values pertaining to ecological agriculture, and those approaches and values that the Andalusian administration and social agents perceive as more functional for their socio-economic and political interests in the short term are imposed. Faced with this scenario, obstacles may persist for the sustainability of Andalusian ecological agriculture, namely, the reproduction of the same economic problems of conventional agriculture, increasing pressure in agro-ecosystem, environmental degradation, and the global interdependencies of organic farmers [86,87]. These include subordination to international markets as the only way to channel local organic production, limited development of local supply chains, submission to the logic imposed by the global agri-food chain, high dependence on multinational producers of seeds and other inputs, agricultural waste management, contamination of underground aquifers, etc. In summary, these socioeconomic circumstances of modernized and industrialized farming have made organic "farmers more dependent on the market and on new technologies to achieve a minimum income threshold; in other words, more dependent on the agro-industrial complex as a whole" [83] (p. 48).

### 5.2. Social and Political Implications

It is likely that influential actors in Andalusian ecological agriculture, immersed in the neo-corporatist logic of the elaboration and development of public policies [73,74], will become agents disconnected from the local rural communities in which the land of organic farmers is located and cultivated. This may become a negative factor for the viability and local sustainability of organic farming, since it can also be subject to the eventual relocation of production and to the changes in land use and land management, which has affected conventional agriculture so much [87,88]. The re-peasantization of Andalusian ecological agriculture in a globalized context of growing interdependencies does not seem plausible,

but socially and environmentally sustainable agricultural practices cannot neglect the interactions and impacts of agricultural agents in their respective territories and local communities [25,27,28]. Nor can it be ignored that these can be the main destination of indigenous organic products, as well as the socioeconomic infrastructure for alternative and local consumption networks. In fact, agricultural production processes are related to eating habits and organic food consumption [82–84,89]. In this way, "the consumption of organic products may be particularly important for institutional marketing in the promotion and rising of awareness towards environmental issues, and in the support to more sustainable forms of consumption [ . . . ]" [89] (p. 201).

The promotion of ecological agriculture that is truly sustainable over time, as well as linked to the local contexts and problems in which it is practiced, requires attending to the so-called illegitimate ecologist approach (semantic field A), which is centered on the paradigm of agroecology and the values of diversity, respect for the environment, and food safety [14,17–19]. However, this requires a profound change in culture and values, new patterns of socialization, and action by the Andalusian ecological agricultural actors, as well as strengthening the perspective of sustainable de-growth [32,33,90]. In addition, the greater or lesser socio-political convergence when defining the priorities of ecological agriculture, between social demands and the government agenda managed by the administration and these actors, will imply a greater or lesser democratization of ecological agriculture. However, it is not evident that greater democratization alone contributes to the social and environmental sustainability of agriculture, due to the conflictive and ambivalent relationship between the ideal of democracy and the various political tendencies of environmentalism [38,39,91,92].

*5.3. New Research Questions*

Other research questions arise that may be useful to deepen the analysis of the specific meanings acquired by the values shown in the three Andalusian plans for organic farming: (1) In what specific linguistic contexts do the different values analyzed appear in the plans, and how do these contexts condition the meanings of these values? (2) What socio-political meanings do the values acquire for the agents who participated in the elaboration of the studied plans? (3) How do the agents who devised the referenced plans justify and/or legitimize the different themes, values, and approaches of ecological agriculture? These new questions require qualitative and ethnographic studies focused on the semantic content analysis and sociological analysis of the discourses of the aforesaid plans (question 1), as well as applying these analysis techniques to possible in-depth interviews and/or focus groups with the agents who are implied in ecological agriculture in Andalusia. The latter would allow a rigorous understanding of the discourses and social representations on the ecological agriculture of these agents (questions 2 and 3) and could be essential to complementing the perspective provided in this article. In any case, studies tackling these research questions could provide a more detailed, localized, and specific description of the themes, approaches, and values on ecological agriculture. Undoubtedly, the new research questions require theoretical approaches and research methods that, on the one hand, are in continuity with those applied in this article and, on the other, introduce some novelties with respect to those developed here [93–95] (see Section 5.4). In short, we think that these can be useful ways to advance the research started in this article.

*5.4. Limitations and Guidelines for Future Research*

The research carried out in this article has different limitations. First, it is the first study carried out from the perspective of content analysis of the Andalusian plans for organic farming. Indeed, up to the present time no research has been published focused on the content analysis of the plans. Although there are numerous studies on the practice and activity of organic farming in Andalusia [16,21,34,35,49,50], from the 2010s to the present, researchers have not systematically studied the contents of the official documents that have planned organic farming in Andalusia in the past two decades. Therefore, it is not possible

to compare the specific results of this study with those of other equivalent previous investigations. However, future studies can replicate the methodology based on the thematic and semantic content analysis used in this article [51–54] to verify whether the themes, approaches, and values studied have continuity in the next Andalusian plans for organic production. As explained in the second and fourth sections of the article, this research applied a qualitative–theoretical sampling in the selection of keywords. The sampling was based on the theoretical arguments and on the data on the evolution of organic farming in Andalusia presented in these sections. This made it possible to systematically study the evolution of the 109 keywords selected for this research, in the form of nouns, in the three plans studied and, thus, to understand the empirical logic of the appearance or absence of the themes, and the approaches and values associated with the former. However, another methodological approach to content analysis and another qualitative sample of keywords, in the form of verbs and nouns, may indicate the emergence of more themes and, probably, new approaches and values [93–98]. For its part, a quantitative content analysis using a representative random probability sample of keywords [99,100] could show the appearance of other themes, as well as other approaches and values different from those studied here.

In addition, the issue of this research has been investigated in a deliberately descriptive way since there are no previous studies on this issue and, therefore, a first appraisal is necessary. The descriptive and localized nature of the research cannot offer explanations about the effective causes for which the themes appear, or not, in the analyzed plans. Specifically, the results obtained in this study, based on a qualitative sample and limited to Andalusian plans, are not generalizable and cannot be extrapolated to organic farming plans published in other regions of Spain and Europe. In this sense, future research could consider the systematic study of the contents of the organic farming plans of other European regions to investigate whether certain themes, approaches, and values are imposed to the detriment of others. The latter is essential to understanding the priorities and social values that have guided public decisions and policies on ecological agriculture in Europe.

Consequently, a relevant issue should be raised in future research in the academic fields of rural sociology, public policy analysis, agrarian social thought, and peasant studies. This issue is that the approaches and social values exposed in the organic farming plans have probably conditioned the agricultural practices and social contexts that organic farmers have experienced in Andalusia and in other European regions. This requires a research effort that cannot be underestimated due to the relevance of such practices and contexts for producers and consumers of organic products, as well as for the sustainability of socio-economic networks and local communities engaged in agricultural activity. This effort requires the preparation of comparative studies and ethnographic and qualitative research for the analysis of the ecological rural environment.

**Author Contributions:** J.-F.J.-D. carried out all the stages of the paper: conceptualization, conception, design, research, analysis, discussion, and conclusions. J.-F.J.-D. and F.C.-C. participated in the final review of article. All authors have read and agreed to the published version of the manuscript.

**Funding:** Not applicable.

**Institutional Review Board Statement:** Not applicable.

**Informed Consent Statement:** Not applicable.

**Data Availability Statement:** The Andalusian plans for organic farming are available at the following official internet link of the Junta de Andalucía: https://acortar.link/cV42z (accessed on 10 January 2020).

**Acknowledgments:** The authors thank the Department of Public Law at Universidad Pablo de Olavide for financing part of the costs of the translation of this article.

**Conflicts of Interest:** The authors declare no conflict of interest.

# Appendix A

**Table A1.** Codification of keywords in Spanish and English.

| Keyword/Number (1–37) | Keyword/Number (38–74) | Keyword/Number (75–109) |
|---|---|---|
| Administración/es [Administration(s)] 1 | Desarrollo local [Local development] 38 | Ordenación [Territorial Planning] 75 |
| Agricultura ecológica [Organic farming] 2 | Desarrollo rural [Rural development] 39 | Organizaciones [Organizations] 76 |
| Agroecología [Agroecology] 3 | Desarrollo sostenible [Sustainable development] 40 | PAC (Política Agrícola Común) [CAP (Common Agricultural Policy)] 77 |
| Agroindustria/s [Agroindustry(ies)] 4 | Desempleados/as [Unemployed] 41 | Participación [Participation] 78 |
| Agua/s [Water] 5 | Desigualdades de género [Gender inequalities] 42 | Participación de las mujeres [Participation of women] 79 |
| Alimentación [Food] 6 | Difusión [Diffusion] 43 | Planificación [Planning] 80 |
| Alimentos ecológicos [Organic food] 7 | Distribución [Distribution] 44 | Política/s [Policy(ies)] 81 |
| Asesoramiento [Counseling] 8 | Diversidad [Diverstiy] 45 | Precio/s [Price(s)] 82 |
| Ayuda/s [Grant(s)] 9 | Divulgación [Dissemination] 46 | Producción ecológica [Organic production] 83 |
| Ayudas públicas [Public aid] 10 | Educación [Education] 47 | Productividad [Productivity] 84 |
| Beneficiarios/as [Beneficiaries] 11 | Empleo (trabajo) [Employment] 48 | Productores [Producers] 85 |
| Beneficio/s [Benefit(s)] 12 | Empresa/s [Company(ies)] 49 | Productos ecológicos [Organic products] 86 |
| Bienestar animal [Animal welfare] 13 | Estrategia/s [Strategy(ies)] 50 | Profesional/es [Professional(s)] 87 |
| Biodiversidad [Biodiversity] 14 | Estrategias de desarrollo local [Local development strategies] 51 | Promoción [Promotion] 88 |
| Cadenas de distribución cortas [Short supply chains] 15 | Europa [Europe] 52 | Protección [Protection] 89 |
| Calidad [Quality] 16 | Experiencia/s [Experience(s)] 53 | Recursos naturales [Natural resources] 90 |
| Cambio climático [Climate change] 17 | Formación [Training] 54 | Red/es [Network(s)] 91 |
| Certificación [Certification] 18 | Género [Gender] 55 | Regulación [Regulation] 92 |
| Ciudadanía [Citizenry] 19 | Gestión [Management] 56 | Rentabilidad [Profitability] 93 |
| Comercialización [Commercialization] 20 | Gran distribución [Large retail] 57 | Residuos [Residues] 94 |
| Competitividad [Competitiveness] 21 | Grupos de Desarrollo Rural [Rural Development Groups] 58 | Respeto al medio ambiente [Environmental protection] 95 |
| Comunicación [Communication] 22 | Igualdad de género [Gender equality] 59 | Responsabilidad/es [Responsibility(ies)] 96 |
| Concentración [Concentration] 23 | Impacto (medio)ambiental [Environmental impact] 60 | Salud [Health] 97 |
| Concentración de la oferta [Supply concentration] 24 | Impacto social [Social impact] 61 | Seguridad alimentaria [Food safety] 98 |
| Conocimiento/s [Knowledge] 25 | Incentivo/s [Incentive(s)] 62 | Sistema de control [Control system] 99 |
| Conocimiento tradicional [Traditional knowledge] 26 | Información [Information] 63 | Sistemas de certificación [Certification schemes] 100 |
| Conservación [Conservation] 27 | Innovación [Innovation] 64 | Sostenibilidad [Sustainability] 101 |
| Consumidor/a/es [Consumer(s)] 28 | Investigación [Research] 65 | Subvención/es [Subsidy(ies)] 102 |
| Consumo [Consumption] 29 | Jóvenes [Young people] 66 | Sustentabilidad [Mantainability] 103 |
| Contaminación/es [Pollution] 30 | Medio ambiente [Environment] 67 | Técnica/s [Technique(s)] 104 |
| Control [Control] 31 | Medio rural [Rural environment] 68 | Tecnología/s [Technology(ies)] 105 |
| Cooperación [Cooperation] 32 | Mercado interior [Domestic market] 69 | Territorio/s [Territory(ies)] 106 |
| Cooperativas [Cooperatives] 33 | Mercado interno [Internal market] 70 | Transferencia [Transfer] 107 |
| Coordinación [Coordination] 34 | Mercado/s [Market(s)] 71 | Transformación [Transformation] 108 |
| Coste/s [Cost(s)] 35 | Mujer/es [Woman/Women] 72 | Transparencia [Transparency] 109 |
| Crecimiento [Growth] 36 | Normas [Norms] 73 | |
| Desarrollo [Development] 37 | Normativa/s [Regulations] 74 | |

Source: Compiled by author. **Note**: Table A1 shows the keywords used for the content analysis of the three Andalusian plans for organic farming. The original language of these plans is Spanish and, therefore, the coding is represented by numbers ordered from lowest to highest according to the alphabetical order of the keywords in the Spanish language. The English translation of the keywords appears in brackets.

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
