# Peer review of "Andalusian Organic Farming Plans (2002–2016): Themes, Approaches and Values"

_sustainability, doi:10.3390/su13063570_

Round 1
Reviewer 1 Report
2021 02 18 Manuscript ID sustainability-1126197
Reviewer Comments
The study explored the most significant production areas and plans for organic produces in the EU – the Andalusian organic farming and plans. The organic agricultural practices in this study had been identified in two versions: ecologists and environmentalists, largely based on the evolution of this agricultural practice in Andalusia during the past decades. Semantic keywords are analyzed to represent the theme of the corresponding versions. The keywords in the government documents of Andalusian organic farming plans are investigated in terms of appearance time and frequencies.
I would like to offer some opinions for the authors’ consideration when they revise this article.
- How and why the documents of the organic plan contains the theme?
It make senses that the theme can be represent by keyword in the plan. However, the reasoning of the local evolution of organic farming and the documentary keywords still needed been explicitly shown in the article.
- This methods might be the foci of the methodology, and it is not enough to present this analysis methods in the section of Method. More comments (pro and con of the methods the authors applied) on the literature should be offered.
- The authors applied the sematic keywords in their analysis. This method should briefly mentioned in the introduction, describe in depth in the section of Theoretical Foundation.
- The authors should addressing much about the “value”, since the tile of this article appear and emphasize it.
- The table titles are too long. The sources, and the notes could be moved below the corresponding table.

Reviewer 2 Report
When reviewing scientific papers for publication, I usually start with a general overview in terms of a structure, abstract, literature review, methodology, findings of the research, discussion, conclusions, as well as limitations of the study and future directions of the research. I also pay attention to the language level, especially if the paper is written in English, and English is not the native language.
The reviewed paper entitled “Andalusian Organic Farming Plans (2002-2016): Themes, Approaches and Values” is generally structured in a proper way. There are, however no sections “Limitations of the study”. These sections should be added too, given this is a research paper.
Abstract. It should be done acc. to the 'from general to details' rule, so first 1-2 introductory sentences, then the purpose of the paper, methodology and finally main findings. There are some introductory sentences and general information about the topic being investigated. In turn, there is neither presentation of the research sample, nor purpose of the study. Key findings of the research are presented, however up to a point only. In addition, one cannot write that something was identified.. (“This research identifies changing perspectives of organic farming developed by the administration and the agents participating in the elaboration of the plans”). If so, you should give more details on it.
Introduction is underdeveloped. The important problem with the paper is that we do not have correctly formulated hypotheses in this paper. Where is the research gap and its justification and hypotheses? There are some general information in this part. Please note that research hypotheses should be formulated in accordance to the following rules: 1) if…. then, or 2) ‘x phenomenon has a positive/negative impact on Y phenomenon. In this case there are just some sentences taken from the literature.
The literature review is quite good and is founded in the existing literature of the topic. Generally I claim that Author (s) provide theoretical foundations for the analysis using appropriate references. I would, however, recommend to add some references to “Discussion” section devoted to the latest literature associated with the topic in question (including Web of Science and Scopus papers).
A very weak point of this paper is "Discussion" section. This section should discuss the results achieved; In addition, there should be references to the results of other scholars. Unfortunately we have not too much in this part, and the second aspect is missing at all. Discussion is an interpretation of the results – implications, significance of results. Provide the response to the research question(s). Interpret results taking into account alternative explanations - where applicable. What are the practical implications (and theoretical –where applicable) suggested by the results of your research. New questions which emerge from your research. Be careful not to “go beyond” your data and results, in particular if the focus of your study is narrow. You can “suggest”, or even “speculate” in the discussion, but it must be clearly evident what is derived from a result and what is your suggestion, comment or speculation, ...
I also recommend a final proofreading of the paper to be done by the native speaker.
Round 2
Reviewer 2 Report
The article has been corrected according to my suggestions. In my opinion, it can be published it is present form.